# Factors Influencing the Implementation of a New Pharmacist Prescribing Service in Community Pharmacies

**DOI:** 10.3390/pharmacy11060173

**Published:** 2023-11-06

**Authors:** Noelia Amador-Fernández, Julie Matthey-de-l’Endroit, Jérôme Berger

**Affiliations:** 1Centre for Primary Care and Public Health (Unisanté), University of Lausanne, 1011 Lausanne, Switzerland; 2Discipline of Pharmacy, Graduate School of Health, University of Technology Sydney (UTS), 2007 Sydney, Australia

**Keywords:** pharmacy-expanded scope, patient care, health care system, pharmacy roles, pharmacy practice, pharmacy education

## Abstract

The pharmacist prescribing service was legally permitted in 2019 in Switzerland to face challenges in the health system; however, there has been a lack of implementation. The aims of this study were to identify implementation factors and to evaluate pharmacy association interventions that aim to support implementation. A qualitative study with two methods was carried out: (1) twelve semi-structured interviews with community pharmacists were recorded, transcribed, and a thematic analysis was carried out using the Consolidated Framework of Implementation Research (CFIR); (2) questionnaires were submitted to the six pharmacy associations of French-speaking Switzerland. The main barriers found were non-reimbursement by health insurance companies, medications’ lack of clinical relevance, a negative perception of GPs, and a lack of time. The main facilitators were the availability of service information, pharmacies belonging to chains/groups, a reduction in the medical consultation burden, and the accessibility of pharmacies. Five associations answered, revealing different initiatives supporting implementation, but none of them had strategies at the political level nor communication strategies aimed at patients or GPs. Based on the CFIR, the most frequent implementation factors were highlighted, and this classification facilitates the transposition of the results to other contexts. The results will allow the development of targeted strategies and add the role of the pharmacy associations, which should be considered in future studies.

## 1. Introduction

Governments and healthcare professionals, such as community pharmacists, share a common goal: improving patients’ clinical outcomes, quality of life and responsible use of medicines [1]. Moreover, healthcare settings around the world are increasingly complex, resource-constrained and interconnected. They are governed by political and economic environments that are equally complex. Optimizing healthcare value has become a policy imperative globally [2]. In this context, research is key for discovering new elements that can contribute to more efficient healthcare. However, proving the effectiveness and efficiency of an innovation does not guarantee that it will be put into widespread use [3], and these discoveries will obviously have no effect if health settings and community pharmacists do not adopt them in daily practice [4].

The transition from evidence-based practices (EBPs) to routine healthcare practice is not spontaneous [2]. According to a literature review aimed at describing and quantifying time lags in the process of applying health research, EBPs take an average of 17 years to be incorporated into routine healthcare practice [5], and only half of EBPs finally reach such routine healthcare practice [2]. This lack of integration of research findings into practice occurs in all settings, disciplines and countries [4]. It is therefore necessary to develop strategies to facilitate the integration of innovations and EBPs into practice in order to ultimately increase their impact on public health. This is the main objective of implementation science [2,3].

In community pharmacies, there is also a growing international trend towards the implementation of services [6]. According to Moullin et al., a professional pharmacy service is defined as “an action or set of actions undertaken in or organized by a pharmacy, delivered by a pharmacist or other health practitioner, who applies their specialized health knowledge personally or via an intermediary, with a patient/client, population or other health professional, to optimize the process of care, with the aim to improve health outcomes and the value of healthcare” [7].

In Switzerland, community pharmacists are an essential pillar in the healthcare system. Over 1800 community pharmacies deal with patients each day. Community pharmacies can be chain pharmacies (38.0%, n = 700), pharmacies included in purchasing communities (32.0%, n = 591), grouped pharmacies with a common market positioning (25.7%, n = 474) or independent pharmacies (4.3%, n = 79). They play a major role in the process of care for chronic and acute patients, but also for people in good health through preventive activities [8]. In some German-speaking cantons, there is no separation between prescribing and dispensing, and general medical practitioners (GPs) are allowed to dispense medications (17 out of 26 cantons); the number of community pharmacies is half compared to those in which dispensing is exclusively performed by pharmacists [9]. This situation decreases access to the healthcare system particularly in rural areas, outside office hours and at weekends [9].

In Switzerland, over the last 15 years, the number of patients who have consulted a GP at least once in the last 12 months has increased by 10%, reaching around 70% of the population [10]. In addition, a study carried out in 2020 estimated that the number of GPs would fall by 16% by 2030 [11]. It is recognized by the Federal authorities that community pharmacists and professional pharmacy services play an important role in relieving the shortage of GPs by increasing patients’ access to primary care [12]. Triage and minor ailment services are services provided by community pharmacists from which patients can receive initial advice for a wide range of health problems. Whitin these services, after the initial pharmacist–patient interview, the pharmacist decides whether to provide non-pharmacological advice, to recommend a medication under his/her own responsibility, or whether to refer the patient to a GP or to emergency services for further evaluation. The Therapeutic Products Act (“Loi sur les produits thérapeutiques” or LPTh) was amended in January 2019 to allow autonomous pharmacists to prescribe (where the pharmacist is solely responsible for the diagnosis, patient assessment and clinical management) [13] treatment for a list of health problems and prescription medications [14]. The current list of health problems which may be managed by pharmacists without a prescription includes seasonal allergic rhinitis, eye disorders, acute respiratory diseases, diseases of the digestive tract, dermatitis, urogenital tract diseases, acute pain, migraine, vitamin and mineral deficiencies, caries prophylaxis, difficulty falling asleep, low blood pressure, travel sickness and dizziness, emergency contraception and smoking cessation. The list of prescription medications includes over 100 medications [14].

Two studies have concluded that the pharmacist prescribing introduced in 2019 is underused in Swiss community pharmacies [15,16]. Pharmacist prescribing is considered an important service by pharmacists for promoting their roles as healthcare providers, relieving the burden of medical consultations and healthcare costs paid by insurances, since patients pay directly for the service. Nevertheless, a study carried out in the German part of Switzerland that aimed to evaluate the factors influencing pharmacists when prescribing medications [15] showed a lack of familiarity with the medications included in the service, pharmacists’ reluctancy to take on the responsibility of prescribing them, and an increased workload when dispensing such medications. Meanwhile, another study that evaluated the clinical relevance of the medications included in the service compared to OTC medications [16] showed that only 19% of the medications were first-line treatments without equivalences in OTC; in addition, it revealed that over 70% of pharmacies included in the study delivered fewer than 50 of the medications included in the service in the previous six months.

The lack of implementation of pharmacist prescribing means that the services’ aims of relieving the burden of medical consultations and healthcare costs [17] cannot be achieved. It is therefore important to conduct research supporting the implementation of such services, as no studies have been carried out in Switzerland to evaluate the implementation factors influencing these services. The aims of this study were, then, to identify the implementation factors (barriers and facilitators) encountered by community pharmacists when implementing the pharmacist prescribing service and to evaluate the strategies undertaken by pharmacy associations to support such implementation.

## 2. Materials and Methods

A qualitative study was carried out in the French-speaking part of Switzerland (Vaud—VD, Fribourg—FR, Jura—JU, Valais—VS, Geneva—GE, Neuchâtel—NE). Two different methodologies were used for the two objectives:Semi-structured interviews deducted and supported by the Consolidated Framework of Implementation Research (CFIR), conducted individually with community pharmacists to identify the implementation factors perceived by pharmacists when implementing the pharmacist prescribing service. The consolidated criteria for reporting qualitative research (COREQ) were used.A questionnaire submitted to the board of pharmacy associations in the French-speaking part of Switzerland to assess their role supporting community pharmacists while implementing the service.

### 2.1. Semi-Structured Interviews

For the semi-structured interviews with pharmacists, the inclusion criteria were licensed pharmacists working in a community pharmacy in the French-speaking part of Switzerland (VD, FR, JU, VS, GE, NE) who had at least one year of experience. The exclusion criteria were a lack of proficiency in French and not giving consent to participate.

Different communication channels were used to recruit community pharmacists. An advert was posted on LinkedIn^®^ and sent to pharmacy associations. After that initial recruitment period, several pharmacies were contacted for intentional recruitment to obtain a certain diversity regarding the type of pharmacy (independent, chain, group), the location of the pharmacy (rural when <10,000 inhabitants or urban when >10,000 inhabitants), and the medications recommended, in line with the pharmacist prescribing service (frequency of use that was considered depending on the number of medications prescribed in the previous 6 months—low when 0–20 prescriptions were made, medium for 21–50 prescriptions and high for over 50 prescriptions—based on a previous study) [16].

To conduct the semi-structured interviews, a guide was developed (Appendix A). The questions included in the interview guide were based on the CFIR classification, which contains 5 domains and 38 factors [18], and on a previous study carried out in 2021 [16]. They were divided into different parts: presentation of the pharmacist/pharmacy (type of pharmacy in which the pharmacist worked, number of people working in the pharmacy, cost of the service); opinion on the pharmacy prescribing service; sources of information and training for the service; time and costs required for implementing the service; implementation protocol; potential patient inclusion criteria for the service and information for patients (promotion); relationship with other pharmacies, relationship with GPs and external incentives; stakeholders involved in implementation; and usefulness of the practice change facilitators (for professional pharmacy services in general). The interview guide was reviewed by two PhD pharmacists with experience in the service (NAF, JEB), and tested by a community pharmacist and PhD student with experience in the service (MEH). Before the start of each interview, the pharmacists were reminded of the objectives and method of the study. The interviews were conducted in the back offices of the community pharmacies in which the pharmacists worked and were all audio recorded. The interviewer was a 2nd-year master’s student in Pharmacy (JME).

For the data analysis, the interviews were transcribed using Trint^®^ v2022 transcription software, then a thematic analysis was carried out manually using MAXQDA Standard^®^ v2022 software. As the CFIR had been used to create the interview guide, it was also used to analyze the transcripts. The interview transcripts were analyzed iteratively, one by one. The various implementation factors (facilitators and barriers) expressed by the pharmacists were identified and classified into a CFIR domain and factor. As the interviews progressed, the number of barriers/facilitators increased. If a barrier/facilitator was unclassifiable in one of the CFIR factors, it was classified under “Other: xxx” in its corresponding domain. To improve reliability, intercoder was carried out by two researchers (JME, NAF) for 10% of the data, which enabled a consensus on the first coding. For the rest of the interviews, a consensus was reached for unclear sections between the two researchers. The number of barriers and facilitators was quantified to identify the most prevalent factors for the implementation of the pharmacist prescribing service.

### 2.2. Questionnaire Given to the Pharmacy Associations

For the second objective, a questionnaire was submitted to the executive committees of the six pharmacy associations in the French-speaking part of Switzerland (VD, FR, JU, VS, GE, NE). It could be completed by the president of the organization or by any other member of the executive committee.

The questionnaire’s aim was to collect any interventions performed by the pharmacy associations to support the implementation of the pharmacist prescribing service in routine healthcare practice. For its development, the websites of the pharmacy associations were consulted, and questions also emerged following initial interviews with pharmacists and discussions between members of the research group. The questionnaire consisted of 5 sections focusing on the following (Appendix A): initiatives in place (i.e., training, provision of documents); communication targeted at patients; communication with GPs; interventions at a political level; and collaboration with the other pharmacy associations. The first four sections included a main question with two conditional questions. The main questions were ‘categorical’ (yes/no); in the case of an affirmative answer, an open-ended question was then asked for respondents to expand their answer, and in the case of a negative answer, an evaluation based on a scale from 1 (totally useless) to 10 (essential) was asked.

The ad hoc questionnaire was reviewed and validated by three PhD pharmacists who were experts in the service (NAF, JEB, ANI) from the research group. The questionnaire was sent via e-mail to the six pharmacy associations concerned in a Word^®^ format. If necessary, up to three reminders were sent via e-mail.

Descriptive analyses were carried out for the categorical questions. Means and standard deviations (SD) were calculated from the numerical responses using Microsoft Excel^®^ v2016. For the open-ended questions, the data obtained were transcribed. Thematic analysis was used to analyze these questions.

### 2.3. Ethics

The protocol was submitted to the Ethics Committee for Human Research of Vaud (CER-VD), who confirmed that the study did not fall within the scope of the Swiss Federal Human Research Act (HRA, RS 810.30) [19] and did not require authorization from an Ethics Committee.

For the semi-structured interviews, participating pharmacists provided written consent to participate after being informed of the study. Regarding the questionnaire for the pharmacy associations, participants were informed of the study via e-mail, and the completion and return of the questionnaire was considered representative of consent to participate.

## 3. Results

### 3.1. Semi-Structured Interviews

Twelve semi-structured individual interviews were conducted with pharmacists working in a community pharmacy in the French-speaking part of Switzerland. Their main characteristics can be found in Table 1. Over 50% of the interviewed pharmacists were female (58.3%) pharmacist owners (58.3%) who worked in an independent pharmacy (50%) in an urban location (58.3%). Regarding the frequency of use of the medications included in the service, the pharmacists interviewed used the medications in equal proportions (33.3%) for high, medium and low use. The individual semi-structured interviews lasted between 20 to 106 min. Only interviews 10 and 11 revealed a new barrier, and no barriers were found in interview 12.

As shown in Table 2, several factors were found for each of the five CFIR domains. Among the 38 CFIR factors, 22 were found to have one or more barrier/facilitator: barriers were found for 19 implementation factors and facilitators were found for 24 factors. Only one new implementation factor had to be added (Other: medications that may be prescribed as part of the service) and it was classified in the domain concerning the “characteristics of the intervention”. Facilitators were more often mentioned by pharmacists than barriers; this was nearly double (496 vs. 272).

The limiting factors most frequently mentioned were non-reimbursement by the mandatory health insurance, the lack of relevance of the medications that may be prescribed as part of the service and/or their dispensing conditions, the negative perception of GPs, and a lack of time to provide the service (Figure 1).

The facilitators most frequently mentioned by pharmacists were information and the knowledge available, being part of a chain or group, relieving the burden of medical consultations, proximity and accessibility (Figure 2).

The five different domains were analyzed depending on the implementation factors expressed by the pharmacists. For the domain “**internal context**”, seven out of the twelve implementation factors included in the classification CFIR [18] were mentioned by pharmacists regarding the implementation of the pharmacist prescribing service. For the structural characteristics of the pharmacies, being part of a chain or group was found to be a facilitator, as it helped by setting prices, offering training, and providing information, etc. Pharmacists believed that “*there are a lot of advantages in the sense that, as it is a big group, there are departments that do certain things. They create a lot of things for us, tools for everyday work*” (Pharmacist 6). Concerning the type of patients, having regular patients was considered a barrier: “*because if it’s a patient we know well, to tell him that he has to sign a document or follow up a new protocol, we say to ourselves that he’s not going to understand and it’s not going to be possible*” (Pharmacist 4). Not having enough staff was another barrier mentioned in relation to the structural characteristics of the pharmacies; “*because if you don’t have enough staff to respond to customers’ needs, you have a team that’s a bit stressed, tired and therefore less attentive. And then, perhaps, if there’s a case where you want to offer a service, discuss it, take it to a consultancy, then, under stress, you may make other choices*” (Pharmacist 2).

Good communication and trust within the pharmacy team was mentioned as a facilitator. It was manifested by the technicians’ confidence in the pharmacists’ skills, which meant that the technicians could easily transfer patients to the pharmacists for the service: “*there’s already a good understanding between us and then there’s confidence in our skills. I think that has an influence. Because as a result, they’ll dare to ask us, or they’ll dare to entrust a patient to us (…). Communication is easy between us, there’s confidence in our abilities. So that’s how it influences us*” (Pharmacist 7).

When preparing for the implementation, management commitment was considered by one pharmacist to a facilitator, because “as head of services, I have to organize all the information, make sure that the team is trained, that the service is in place, that there is proper communication between the team and outside, and that the prices are appropriate” (Pharmacist 2). With regard to the information available about the service, one pharmacist said that “*we had a lot of information already but now they’re (associations and researchers) also starting to develop a lot of information with a focus on community pharmacy*” (Pharmacist 3). However, the available resources were frequently deemed a barrier, as “IT tools are not yet fully developed for the pharmacist prescribing service” (Pharmacist 8), and time is needed when dispensing a prescription medication; this was considered to take longer than dispensing an OTC. Pharmacists said that the time available was linked to staff resources.

For the domain “**characteristics of individuals**” (pharmacists and pharmacy staff’s characteristics), four out of the five implementation factors included in the classification CFIR [18] were mentioned by pharmacists regarding the implementation of the pharmacist prescribing service. Regarding pharmacy staff’s knowledge and beliefs about the service, pharmacist prescribing was frequently considered to help reduce the medical consultation burden: “*there is a lot of talk about the shortage of GPs (…). Putting more value on the work of pharmacists and giving us more responsibility is a very good choice, because pharmacies are everywhere. And the number of times that, now that we can dispense this type of medicine, we’ve avoided trips to emergency departments and nights in those departments for people who first went to the pharmacy, I think that’s really good. It’s going to make… At least we hope so, save the health system a lot of money*” (Pharmacist 5). They see the service as an advantage for patients due to the proximity and accessibility of pharmacies: “*I think it’s good for the customer, because pharmacy distribution is already fairly accessible, with extended opening hours*” (Pharmacist 2). Pharmacists also consider the service to be an opportunity for promoting pharmacists’ work and its integration in primary care: “*The major advantages are that it gives pharmacists the tools they need to manage common pathologies. That’s the idea. So, in the end, we’re getting away from the principle that we’ve always fought against, of being drug sellers. And that gives us a boost in terms of being recognized as a healthcare provider. A very good idea*” (Pharmacist 3).

On the other hand, an implementation factor often considered to be a barrier was the lack of training for pharmacy staff (pharmacists and technicians): “*I’m not going onto a slippery slope just yet, because I think it’s something we would have to do if we had the training to do it*” (Pharmacist 11); “*The aim of training technicians is to ensure that they know that these drugs can be delivered, and that they transfer the patient to us (pharmacists). So, for me, training isn’t more important than having the theoretical basis of Pharma-News (publication targeting information for technicians) or having an idea of what’s done in medication list*” (Pharmacist 7).

For the domain “**characteristics of the intervention**”, three out of the eight implementation factors included in the classification CFIR [18] were mentioned by pharmacists regarding the implementation of the pharmacist prescribing service. As mentioned before, a new implementation factor was added in this domain regarding the medications that may be prescribed as part of the service. This factor was repeatedly considered as a barrier; pharmacists felt that some of the medications were not clinically relevant or did not offer any added value compared to the medications available as OTCs. In addition, the dispensing conditions were often seen as a disadvantage: “*There are sometimes restrictions, triptans are on the list, but the restriction is such that you already need a previous prescription from your GP. So, in the end, we don’t add much value by reintroducing a triptan that has already been prescribed*” (Pharmacist 4). Another barrier often mentioned was the non-reimbursement of the service by the compulsory health insurance: “*The only disadvantage is that they (patients) have to pay out of pocket. That’s complicated. In any case, in a local pharmacy, we do it, but it’s still done very sparingly*” (Pharmacist 6). However, as a facilitator for this domain, some pharmacists mentioned that patients value the service provided (professionalism and improved access) over its costs (lower prices than for GP consultations): “*There are plenty (of patients) who are prepared to pay out of their own pocket to avoid having to wait to go to the GP, to have a solution right away*” (Pharmacist 7).

For the domain “**external context**”, three out of the four implementation factors included in the classification CFIR [18] were mentioned by pharmacists regarding the implementation of the pharmacist prescribing service. One of the main barriers mentioned by pharmacists was the negative perception of GPs regarding the service; some said that GPs feel that pharmacists are taking over their patients, and that this thought is more prevalent among older GPs. This barrier has an important impact on the way the service is advertised and/or implemented: “*Some GPs don’t like the idea of us recommending their patients this kind of medication. So, it’s true that, before doing a recommendation, I check the patient’s GP, knowing that sometimes it’s better that I don’t recommend prescription medication*” (Pharmacist 12).

On the other hand, some pharmacists believed that pharmacy associations at the national or local level provide support for the implementation of the service: “*I take a fairly positive view of it. I think we have a very good association, pharmaSuisse, which gives us a lot of support. I’ve seen a lot of reports and articles on the subject, and they’re very encouraging in that sense. It’s stimulating, it allows us to say, ‘come on, we’ve got support’. I think it’s very positive*” (Pharmacist 1).

For the domain “**process**”, two out of the nine implementation factors included in the classification CFIR [18] were mentioned by pharmacists regarding the implementation of the pharmacist prescribing service. The barrier most frequently mentioned was patients’ lack of awareness about professional pharmacy services, such as the pharmacist prescribing service: “*Patients aren’t necessarily aware of this possibility (pharmacist prescribing service), so when I suggest it to them, they’re generally pleased*” (Pharmacist 11). Pharmacists thought that the general public was unaware of the pharmacist’s role as a healthcare provider and that it would be useful to inform them of this; some mentioned that this should be organized in their pharmacy or by the pharmacy associations.

### 3.2. Questionnaire to the Pharmacy Associations

Five out of the six pharmacy associations included in the study answered the questionnaire provided.

Regarding the initiatives in place to support the implementation of the pharmacist prescribing service, all the associations (n = 5) had strategies to help pharmacists integrate the service into routine healthcare practice. These initiatives were as follows: training for pharmacy staff (n = 4), standardization of the list of services provided (n = 1), encouraging pharmacists to attend training about primary care anamnesis (n = 1), raising awareness about the service among pharmacy technicians (n = 2), and offering documentation to help provide the service (n = 3). Only one of the associations answered that no training was available, but rated its importance at a 9 out of 10 (10 being essential).

None of the associations were found to have implemented a communication strategy to inform patients about the pharmacist prescribing service available in community pharmacies. The mean rated importance of implementing such communications was 6.4 out of 10 (SD = 2.7). One of the associations stated that such a strategy could help the service; however, it could also result in a negative reaction from GPs and, therefore, it would be more appropriate for pharmacies to implement a “*discreet and individual communication strategy*”.

None of the associations was found to have implemented a communication strategy to inform GPs about the pharmacist prescribing service. The mean rating for the importance of this kind of communication was 4.0 out of 10 (SD = 2.9). One of the associations affirmed that disseminating information between community pharmacists and their surrounding GPs would be more effective than a general communication strategy from the association. Another association considered the strategy important; nevertheless, it should be targeted depending on the medical density surrounding the community pharmacy and the relationship between the pharmacists and GPs. A third association felt that support from GPs should not be expected; however, it would still be important to inform them through GP–pharmacist communication circles [20].

None of the associations were found to have implemented a communication strategy to inform the political level. The mean rating for the importance of a political strategy was 4.8 out of 10 (SD = 2.4). The associations that considered the strategy to be less important stated that the legal framework already existed, so now it is up to the pharmacists to put it in place.

The majority of respondents (n = 3) declared that they had not collaborated with the National Pharmacists Association (pharmaSuisse) and/or other associations. The respondents who had collaborated (n = 2) with this association had provided training and held meetings in collaboration with pharmaSuisse about the pharmacist prescribing service.

## 4. Discussion

To our knowledge, this is the first study to identify and classify the main barriers and facilitators when implementing a pharmacist prescribing service. The use of the CFIR classification facilitates the transposition of the results to other contexts and countries; in fact, only one new factor had to be added to the CFIR. The barriers most frequently mentioned by pharmacists were the non-reimbursement of the service by the compulsory health insurance, the lack of relevance of the medications included in the service compared to OTCs or depending on their dispensing conditions, the negative perceptions of GPs, and pharmacists lacking time to provide the service. In terms of facilitators, the most frequently cited were the availability of information and knowledge relating to the service, the inclusion of the pharmacy in a chain or group, the advantage of the system regarding its ability to relieve the burden of medical consultations, and the advantage for patients due to the proximity and accessibility offered by the service.

The implementation factors most frequently thought to hinder the integration of the service into daily practice were related to three domains (external context, internal context and characteristics of the service). The non-reimbursement of the service by the health insurance was a factor often mentioned by the pharmacists as limiting the implementation. Payment for services is another barrier often mentioned when implementing services, as shown by Gastelurrutia et al. [21]. A study carried out in Switzerland concerning the pharmacist prescribing service also stated that pharmacists considered this limitation “*very important*” [16]. Another study carried out at the University of Basel highlighted this point [15]; it added that patients disapproved of the cost of the service and that appropriate communication about the price could be an appropriate strategy for overcoming such a barrier. Negotiations with health insurers will have to be undertaken before the service can be reimbursed, and the National Pharmacist Organization may play a key role in this. pharmaSuisse has suggested that “*gathering reliable data to demonstrate that consultations in pharmacies can reduce the number of emergency room visits*” is needed [22]. Over the past ten years, the Swiss population has changed its choice of deductibles. This change is particularly noticeable for the highest deductible (CHF 2500), since the proportion of policyholders preferring this deductible has doubled, rising from 15% in 2007 to 29% in 2017 [23]. For these people with a high deductible, the fact that the service is covered by the mandatory health insurance will not change anything; the patients will pay out of pocket. However, reimbursement would allow a standard price to be set for the service, as at the moment, each pharmacist must fix the service fee for his/her own pharmacy.

Many pharmacists believe that GPs have a negative perception of the new responsibilities entrusted to pharmacists. This factor has already been studied as a barrier for other professional pharmacy services [21].I In addition, it has been perceived as a barrier also by patients. Indeed, a study looking at potential users’ views on pharmaceutical benefits found that some participants perceived it as an intrusion into the competencies of other healthcare professionals [24]. Communicating with GPs would help to overcome this barrier, as well as the barrier concerning “GPs’ lack of knowledge of the service”. Communication would help to clarify the pharmacist’s role in the healthcare system. Given that the associations have not communicated about this service to GPs, it would be important to evaluate their real knowledge about the service in a further study to determine whether the barrier is real or whether it is only perceived by the pharmacists.

A lack of time was another major barrier; this has also been shown in studies that aim to evaluate the implementation of professional pharmacy services [21,25,26]. For the pharmacist prescribing service, more time is needed for consultation and the documentation of the intervention compared to the regular dispensing of medications. In addition, time is needed to put in place any new service in the pharmacy. This barrier was influenced by other identified factors, such as staff shortages, but also an unsuitable IT system or a lack of experience with the service, which are both caused by lower efficiency. Another barrier, which was certainly linked to a lack of time, was motivation. Motivation increases personal and/or team commitment, and encourages investment in resources, as highly motivated pharmacists will certainly give greater priority to implementing the service. In fact, a lack of motivation is frequently described as a barrier in the literature [26,27,28], although the presence of motivation in champions of the pharmacy service can be considered a facilitator to overcome challenges when implementing services [29]. Therefore, it is possible that a lack of time is caused by other factors, as shown in another study [30], where a lack of time was “the most critical implementation factor”, and it was linked to many factors such as workflow. A previous study [31] categorized the strategies most frequently used to overcome barriers such as a lack of time: adapting the distribution of tasks, adapting the vision/mission, creating a timetable, etc. These are strategies that have been considered elsewhere for the implementation of professional pharmacy services [32].

Other barriers were also mentioned in the study. Some pharmacists felt that their patients were unaware of pharmacists’ role as healthcare providers. This perception was also shown in another study where it was noted that several participants were unfamiliar with the pharmacy service studied, and that some patients perceived pharmacies “as perfumeries rather than healthcare establishments” [24]. Related to the patients’ beliefs, one study [28] identified that the insufficient promotion of a service is a barrier to its implementation. A different study also mentioned that community pharmacists see themselves as “drug dispensers” rather than “patient-centered practitioners”, which poses a barrier to the adoption of new professional pharmacy services [33].

In relation to the facilitators, the implementation factors most frequently thought by pharmacists to facilitate the integration of the service into daily practice were related to two domains: internal context and the characteristics of the pharmacy staff. Access to information and knowledge within the internal context of the pharmacy have been considered “important to ensure that pharmacists have the necessary skills or training to provide the professional service” [34], and it has also been reported in previous studies as a facilitator for the implementation of services [35].

Being part of a chain and/or group of pharmacies was also considered an enabler for the implementation of the pharmacist prescribing service. Another study [35] also mentioned that affiliation to a chain facilitated the implementation of professional pharmacy services. In fact, it reported that “a significantly higher number of medication reviews were carried out in UK pharmacy chains than in independent pharmacies”. In the present study, among the pharmacies interviewed, none of those belonging to a chain had a low frequency of using the service. The chains’ central office can facilitate changes in practice by providing the necessary equipment, organizing training, implementing a commercial strategy, and defining prices for the service, etc. In addition, they set the targets that responsible pharmacists are accountable for, while independent pharmacies do not necessarily have them. The groups also helped pharmacies to implement new services, but to a lesser extent.

Regarding the characteristics of pharmacists, they strongly believed that a major advantage of the service within the health system was its capacity for reducing the burden of unnecessary medical consultations. In a previous study about the implementation of a community pharmacy anticoagulation management service [25], it was noted that a decrease in the burden of medical consultations led to a reduction in the workload of GPs. Pharmacists also believed that the proximity and accessibility of community pharmacies was a major facilitator for patients, and could help to integrate the service in routine healthcare practice. “Reduction in medical consultation burden” and “proximity and accessibility” were linked to the factor “knowledge and beliefs about the intervention”, which is included in the domain “characteristics of individuals” of the CFIR. Only facilitators that could offer benefits to pharmacists, patients and the healthcare system were considered.

The patient characteristics that were also considered facilitators for the service were irregular patients and patients without a GP and/or with a high deductible. The results of a study carried out in Switzerland were also in accordance with this, and it was mentioned that the service was particularly interesting for “people who are professionally active and have no chronic illness” [15]. Practicing as a pharmacist in an environment with a low density of GPs (e.g., rural area) was also mentioned as an enabler, since the demand for primary care is higher.

As for future studies and initiatives, the development of targeted strategies to overcome the barriers found in the present study and to maintain the facilitators in order to better integrate the service in daily pharmacy practice is needed. For that same reason, it was important to evaluate the strategies already put in place by the pharmacy associations. The results showed that pharmacy associations in the French part of Switzerland have initiatives related to training, documentation, encouragement, etc.; however, they have not put in place strategies for communicating with patients, GPs, nor politics, as they generally suggest that it is preferable for each pharmacy to communicate personally with the surrounding professionals. During the interviews with the pharmacists, some of them suggested that it was up to the associations to implement a marketing strategy to inform patients about the services provided by pharmacists and their role as service providers. Without this type of support from the associations, pharmacists must adopt a “pull” marketing strategy (e.g., the pharmacist has to propose/present the service to the patient, convincing him/her that it is useful). With such communication strategies, the pharmacists would be moving towards a “push” marketing strategy, which is closer to the usual practice of pharmacies (patients would come to ask for a service/product with an expectation) [36].

An association answered that its aim was to standardize the services offered by community pharmacies in its region. This would compensate for a barrier found in the external context domain regarding the heterogeneity of practices in the different pharmacies. However, this would require training, support, tools, etc., and, therefore, a lot of resources. This should also be performed at the national level, rather than at the local level.

## 5. Strengths and Limitations

As strengths, the variety of pharmacists included in this study regarding their use of the service, the type of pharmacy and the geographical context (village/town and regions) enabled different experiences and opinions to be gathered. Saturation could also be confirmed, as one or no new codes were obtained for the last three interviews (interviews 10 and 11 only showed a new barrier, and no new barriers were found in interview 12). In addition, the survey given to the pharmacy associations complemented the semi-structured interviews including five out of the six associations, and enabled the results obtained to be put into context and their interpretation to be strengthened. In terms of analysis, intercoder reliability was evaluated to improve the systematicity, communicability, and transparency of the coding process [37].

As for the study limitations, the interview guide and the analysis were not based on the latest version of the CFIR [38]. An updated version was published in October 2022 when the interviews and their analysis had already been carried out. However, although new factors have been added to the new version, the five domains remain the same. In addition, a critical review of the literature that identified the most relevant CFIR factors for the implementation of different services in community pharmacies determined that several factors have not been observed in the literature, including the factor “targets and feedback” [31].

The number of barriers/facilitators was used to help with the presentation of the data. Given the qualitative nature of this research, these values have no significance regarding the identification of the main barriers or facilitators for such a service (this would require a quantitative study that could be based on these results).

## 6. Conclusions

The aims of this qualitative study were to identify the barriers and facilitators affecting the implementation of a pharmacist prescribing service in the French part of Switzerland, and to identify the roles of the pharmacy associations in supporting the implementation. Pharmacist prescribing is an important service that can increase pharmacists’ responsibilities and their integration into primary care. Therefore, its implementation must be enhanced to achieve better outcomes for patients and health systems. Identifying barriers and facilitators is the first step towards improving the integration of the service into pharmacy practice, adding the role of pharmacy associations, groups and chains; this should be considered in future studies when evaluating the implementation of professional pharmacy services.

Based on the CFIR, the most frequent implementation factors were highlighted. This classification facilitates the transposition of the results to other countries; therefore, the present results can help to facilitate the implementation of prescribing services in community pharmacies in different contexts. This will also enable targeted strategies to be developed in order to improve implementation.

## Figures and Tables

**Figure 1 pharmacy-11-00173-f001:**
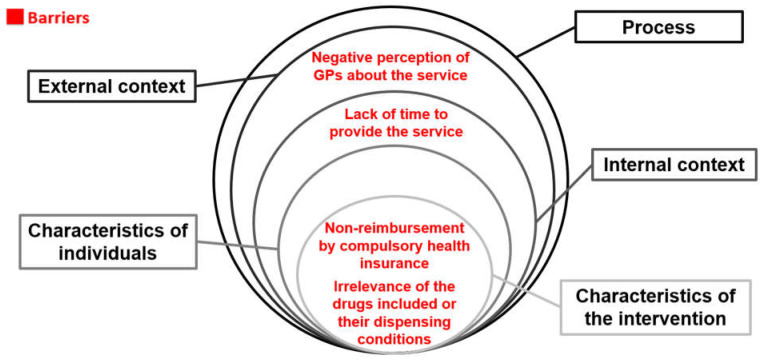
The most prevalent barriers mentioned by community pharmacists when implementing the pharmacist prescribing service according to the CFIR.

**Figure 2 pharmacy-11-00173-f002:**
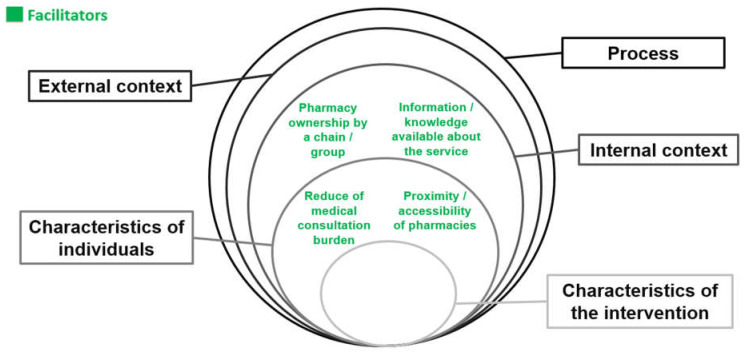
The most prevalent facilitators mentioned by community pharmacists when implementing the pharmacist prescribing service according to the CFIR.

**Table 1 pharmacy-11-00173-t001:** Participant pharmacists’ characteristics.

N°	Gender	Pharmacist Position	Type of Pharmacy	Location *	Canton	Work Rate (%)	Diploma Obtention (Year)	Pharmacy Staff: Pharmacists	Pharmacy Staff: Technicians	Frequency of Use **
1	Male	Owner	Independent	Urban	NE	80	2009	2	4–6	Low
2	Female	Staff	Independent	Rural	NE	60	2019	2–4	6–7	Low
3	Male	Owner	Independent	Rural	VS	100	2009	2–3	6–7	High
4	Female	Co-owner	Independent	Rural	VD	100	2009	3	5	Medium
5	Female	Staff	Independent	Urban	GE	40	2019	5	5	High
6	Female	Responsible	Chain	Urban	VD	100	2019	1–2	3	Medium
7	Male	Co-responsible	Group	Urban	VD	85	1999	2	6	Medium
8	Male	Owner	Independent	Rural	VD	100	1989	2	2	Low
9	Female	Owner	Chain	Urban	VD	85	2019	1–2	3	High
10	Male	Owner	Chain	Urban	FR	100	1989	2	5	High
11	Female	Owner	Group	Rural	GE	90	2008	1	1	Low
12	Female	Staff	Chain	Urban	FR	100	2018	2	4	Medium

* Rural < 10,000 habitants; Urban > 10,000 habitants. ** Frequency of use: number of medications prescribed in the previous 6 months (low when 0–20 prescriptions, medium when 21–50 prescriptions and high when over 50 prescriptions) [16].

**Table 2 pharmacy-11-00173-t002:** Domains and implementation factors mentioned as barrier or facilitator in the interviews according to the CFIR classification.

Domains	Factor	Barriers	Facilitators
Internal context	Internal characteristics	--	Pharmacy owned by a chain and group (n = 43)
		Having regular patients (n = 7)	Having walk-in patients (n = 4)
		Lack of staff (n = 12)	Adequate number of staff (n = 8)
	Network and communication	--	Communication and trust within the pharmacy team (n = 23)
	Culture	--	Professional pharmacy services set as an objective in the pharmacy (n = 2)
	Implementation climate—Objectives and feedback	Absence of set objectives (n = 8)	Presence of set objectives (n = 7)
	Preparing for the implementation—Management commitment	--	Coordinator for the implementation (n = 21)
	Preparing for the implementation—Available resources	Inadequate IT system (n = 14)	Adequate IT system (n = 11)
		Insufficient time available for the service (n = 16)	Sufficient time available for the service (n = 10)
	Preparing for the implementation—access to knowledge and information	Lack of compulsory training related to anamnesis (n = 5)	--
		Available training considered as insufficient or incomplete (n = 4)	Available training considered as relevant, of high quality (n = 5)
		--	Information and knowledge available (n = 51)
		Lack of targeted information about the service (n = 13)	Adequate targeted information about the service (n = 2)
Characteristics of individuals (pharmacy staff)	Knowledge and beliefs about the service	--	Reduce of medical consultation burden (n = 36)
		--	New system of remuneration for pharmacies (n = 9)
		--	Proximity and accessibility of pharmacies for patients (n = 33)
		--	Pharmacists’ job valorization and integration of pharmacies into primary care (n = 20)
		--	Improvement in patient loyalty (n = 1)
	Self-efficacy	Lack of knowledge and/or skills (n = 11)	Adequate knowledge and/or skills (n = 16)
		Lack of pharmacists’ self-confidence (n = 9)	Adequate pharmacists’ self-confidence (n = 2)
	Individual stage of change	Lack of understanding from the team regarding the challenges of the service (n = 2)	--
		Pharmacists not trained (n = 15)	Pharmacists adequately trained (n = 19)
		Technicians not trained (n = 14)	Technicians adequately trained (n = 13)
	Other personal characteristics	Lack of experience (n = 3)	Adequate experience (n = 15)
		Lack of motivation (n = 6)	Presence of motivation (n = 4)
Characteristics of the intervention (pharmacist prescribing service)	Complexity	Complexity of the service (n = 7)	--
	Quality and presentation of the intervention	--	High quality of the information included on the FOPH (Federal Office of Public Health) website (n = 3)
	Patients’ costs	Non-reimbursement by compulsory health insurance (n = 42)	--
		--	Added value of the service compared with costs (n = 20)
		Inappropriate communication about the price of the service (n = 8)	Appropriate communication about the price of the service (n = 15)
	Pharmacies’ costs	--	Low and justified investment costs (n = 14)
	Other: medications included in the service	Irrelevant medications and/or delivery conditions included in the service (n = 29)	Relevant medications and/or delivery conditions included in the service (n = 13)
External context	Patient needs and resources	Patients with doctor and/or low deductible (n = 7)	Patients without doctor and/or high deductible (n = 10)
		Insufficient patients to recommend the service (n = 3)	Sufficient patients to recommend the service (n = 12)
	Network with GPs	GPs’ lack of awareness of the service (n = 2)	GPs’ awareness of the service (n = 3)
		GPs negative perception of the service (n = 18)	10
	Network with other community pharmacies	Heterogeneity of practices (n = 5)	--
		--	Collaboration to set service prices (n = 9)
	Political or external incentives	Lack of access to patients’ history files (n = 2)	--
		--	Support from the pharmacy association (n = 19)
Process	Planification	Service fee non-fixed (n = 3)	Service fee fixed (n = 9)
	Implication of patients	Patients’ lack of knowledge about the pharmacists’ role as care providers (n = 7)	Patients’ knowledge about the pharmacists’ role as care providers (n = 4)
5 Domains	22 Implementation factors	27 Barriers mentioned	36 Facilitators mentioned

## Data Availability

Data will be provided upon requirement.

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
