# Peer review of "Factors Influencing the Implementation of a New Pharmacist Prescribing Service in Community Pharmacies"

_pharmacy, 2023, doi:10.3390/pharmacy11060173_

Round 1

Reviewer 1 Report

Comments and Suggestions for Authors

Thank you for the opportunity to review this manuscript ‘Factors influencing the implementation of a new pharmacist 2 prescribing service in community pharmacies’. Minor revision is recommended.

Introduction

It would be worthwhile to provide an overview of roles of community pharmacists in Switzerland and more details about situations where GPs can dispense medications, as this type of arrangement may be less common in other countries with separation of dispensing and prescribing.

Is limiting medical consultations one of the benefits of pharmacist prescribing? Or is relieving the burden of high demands for medical consultations is more appropriate?

Line 86 - lack of willingness to take responsibility from pharmacists is a bit vague. Does it mean that pharmacists were reluctant to take on the responsibility to diagnose and prescribe medications included in this service? More details are required to provide clarity.

Line 91 – Please expand on the cost-effectiveness of pharmacist prescribing as this is one of the rationales to support the conduct of this study.

Line 81 – How many studies have been conducted to assess the frequency of pharmacist prescribing in Switzerland. If there were only two studies (Ref 15 and 16) then please expand on the findings and provide some figures to highlight the need for this study to be conducted.

Methods

Line 98 – It is stated that the study was conducted in the French speaking part of Switzerland and there was a study conducted in the German part of Switzerland (Ref 15). It would be good to mention this in the introduction to provide the readers with more context as to why this study was conducted e.g. no previous study evaluated the pharmacist prescribing services in this part of Switzerland.

Results

Participants’ characteristics should be summarised and briefly described in the results section, in particular to outline the frequency of use of pharmacist prescribing services.

Line 269 - Similar to comment above, avoid medical consultation should be reworded to e.g. ‘reduce medical consultation burden’

Line 289 – This sentence needs to be revised. The sentence is unfinished neither pharmacist nor technicians …?

Table 2

The themes are too brief to provide context. It would be helpful to have more specific themes e.g. ‘lack of staff’ as a barrier and ‘adequate staffing’ as a facilitator to provide clarity.  The number of study can be included in brackets next to the subthemes e.g. lack of staff (n=12)

How does ‘Lack of compulsory training related to anamnesis’ differ to ‘Insufficient or incomplete training’?

Suggest to change anamnesis to history taking.

Avoidance of medical consultations. This theme is unclear and needs to be reworded.

Strengths and limitations

Line 513 – one or no theme is confusing, suggest to change to ‘only one new theme emerged in the last three interviews, with no new theme obtained in the last interview’.

‘/’ was used to differentiate between barriers and enablers. However, it looks like ‘Pharmacists’ job valorization/Integration in primary care’ is a facilitator theme, even with ‘/’ included. Therefore categorising the themes into the barriers column and the facilitators column is recommended.

FOPH website – more details are required. Was it the ease of use of the website?

I wonder how negative perception of GPs about the service is considered as a barrier and a facilitator?

Discussion

It would be good to compare some of findings from this study to Ref 15 in the discussion.

Line 409 - However, it would allow to set a price and justify that it is something that  needs to be paid. This sentence is unclear.

Given that negative perception of GPs about pharmacist prescribing is a key barrier, were there any previous studies exploring GPs’ views on pharmacist prescribing which could be included in the discussion?

It’s unclear how lack of motivation can be a facilitator to implement a service (Ref 28), more explanation is required.

Mechanics

English language revision is required. Suggested revisions are outlined below.

Line 67 –  Witin, should be within.

Line 77 – migraine crises, consider just migraine or migraine attacks.

Line 85 – there is ‘a’ lack of

Line 121 – prescription.s, remove .

Line 158 – was collecting, suggest to change to was to collect

Line 162 – Suggest to remove regarding structure

Line 167 and 168 – In case affirmative/negative, suggest to change to in cases of affirmative/negative answers

Line 170 – review for validation, was it reviewed and validated

Line 170 – PhD pharmacists (who are) experts ….

Line 229 – are should be were analyzed

Line 246 – A good communication, a is not needed here

Line 279 – pharmacist(s’) work

Line 285 – On (in) the other hand

Table 2 – within not withing

Line 454 – an “important” – should it be as?

Line 516 – enables(d)

Line 517 – consider changing to ‘In terms of analysis, intercoder reliability was evaluated, as a good practice in qualitative analysis, to improve systematicity, communicability, and transparency of the coding process.

Comments on the Quality of English Language

Extensive editing of English language is required.

Author Response

Thank you very much for the time and thorough review, we have answered each one of the comments separately in the table below.

Reviewer 2 Report

Comments and Suggestions for Authors

Excellent paper and well presented application of CFIR framework. No changes recommended.

Author Response

Thank you very much for the time and thorough review.

Reviewer 3 Report

Comments and Suggestions for Authors

This is a well written paper based on a sound and fundamentally appropriate research design. The results of the research and paper will allow for implantation of actions to increase the prescribing capacity of pharmacists. 

Author Response

(The authors gave the same response as above.)

Reviewer 4 Report

Comments and Suggestions for Authors

Thank you for the opportunity to review the manuscript "Factors influencing the implementation of a new pharmacist prescribing service in community pharmacies"

The topic is interesting and relevant for the development of pharmacy today. However I have some considerations, mostly related to methods.  

I find the introduction relevant as it takes into consideration not only the specific intervention of pharmacist prescribing in Switzerland, but also the wider topic of implementation. However, it lacks something about pharmacist prescribing in other countries. As for now it moves directly from pharmacy services in general to the specific case of Switzerland (line 53) without touching on pharmacist prescribing in the wider context.

It would make it easier for the reader to insert a sub-heading like “The context” or similar before starting the description of Switzerland – or at least make it a new paragraph.

Material and methods

The authors state that the interviews were based on grounded theory. I cannot see in what sense they were based on that. On the contrary they are based on a specific model, and the questions in the interview guide are rather detailed – pointing to the opposite of grounded theory. At least explain in what way grounded theory was used and add a reference.

Not only the interview guide, but also the analysis seems to have been deductive using the CFIR frame this is not clearly stated.

About the interview guide, there is a risk that it leads to biased answers. For example: “Do you feel that you have enough time…” is very different from asking “what challenges have you experienced…” The first points the respondent to think about time, whereas a more open question would make them think more freely from their own experience. In all this seems more like a quantitative approach (but with a limited sample).

Not only was the interview guide was created using CFIR, but also the analysis seems to be deductive using the CFIR frame, which seems to be less exploratory than you would expect from a grounded theory (or any qualitative) approach.

Sample. It is stated that “intentional recruitment” was used. What does this mean? What was the sampling strategy?

In general, for the qualitative part, please look at the COREQ guidelines (for reporting qualitative data) and make sure all relevant items are addressed.

For the questionnaire – how were data gathered from open questions arranged/analyzed?

Results

In the result section a lot of the data is quantified. This is something to reflect on in qualitative studies, especially considering what sampling strategy is used. The reasons for quantifying needs to be described or reflected on, either in method or in methods discussion.

Minor comment: Figure 1 is rather busy, consider taking the quotes out.

The results section is descriptive, i.e. reporting each item in a superficial (not-qualitative) way. Could it be reported more in-depth (maybe needing a re-analysis with more interpretation)?

Discussion

The discussion contains much repetition of results, and could hence be shortened. It could be more concentrated on the interpretation of the results.

There is also a need for more of methods discussion/reflections.

Author Response

(The authors gave the same response as above.)
